# Effects of ε-Poly-L-Lysine/Chitosan Composite Coating on the Storage Quality, Reactive Oxygen Species Metabolism, and Membrane Lipid Metabolism of *Tremella fuciformis*

**DOI:** 10.3390/ijms26157497

**Published:** 2025-08-03

**Authors:** Junzheng Sun, Yingying Wei, Longxiang Li, Mengjie Yang, Yusha Liu, Qiting Li, Shaoxiong Zhou, Chunmei Lai, Junchen Chen, Pufu Lai

**Affiliations:** 1Institute of Food Science and Technology, Fujian Academy of Agricultural Sciences, Fuzhou 350003, China; sunjzll@163.com (J.S.); wyy787459024@163.com (Y.W.); 13616901013@163.com (L.L.); yummy2023yummy@163.com (M.Y.); yy613666@163.com (Y.L.); lqt6868@163.com (Q.L.); zhousx2023@163.com (S.Z.); 19911583175@163.com (C.L.); junchenccc@163.com (J.C.); 2National R&D Center for Edible Fungi Processing, Fuzhou 350003, China; 3Key Laboratory of Subtropical Characteristic Fruits, Vegetables and Edible Fungi Processing (Co-Construction by Ministry and Province), Ministry of Agriculture and Rural Affairs, Fuzhou 350003, China; 4College of Food Science, Fujian Agriculture and Forestry University, Fuzhou 350002, China; 5College of Life Sciences, Fujian Agriculture and Forestry University, Fuzhou 350002, China

**Keywords:** *Tremella fuciformis*, ε-Poly-L-lysine, chitosan, reactive oxygen species metabolism, membrane lipid metabolism

## Abstract

This study aimed to investigate the efficacy of a composite coating composed of 150 mg/L ε-Poly-L-lysine (ε-PL) and 5 g/L chitosan (CTS) in extending the shelf life and maintaining the postharvest quality of fresh *Tremella fuciformis*. Freshly harvested *T. fuciformis* were treated by surface spraying, with distilled water serving as the control. The effects of the coating on storage quality, physicochemical properties, reactive oxygen species (ROS) metabolism, and membrane lipid metabolism were evaluated during storage at (25 ± 1) °C. The results showed that the ε-PL/CTS composite coating significantly retarded quality deterioration, as evidenced by reduced weight loss, maintained whiteness and color, and higher retention of soluble sugars, soluble solids, and soluble proteins. The coating also effectively limited water migration and loss. Mechanistically, the coated *T. fuciformis* exhibited enhanced antioxidant capacity, characterized by increased superoxide anion (O_2_^−^) resistance capacity, higher activities of antioxidant enzymes (SOD, CAT, APX), and elevated levels of non-enzymatic antioxidants (AsA, GSH). This led to a significant reduction in malondialdehyde (MDA) accumulation, alongside improved DPPH radical scavenging activity and reducing power. Furthermore, the ε-PL/CTS coating preserved cell membrane integrity by inhibiting the activities of lipid-degrading enzymes (lipase, LOX, PLD), maintaining higher levels of key phospholipids (phosphatidylinositol and phosphatidylcholine), delaying phosphatidic acid accumulation, and consequently reducing cell membrane permeability. In conclusion, the ε-PL/CTS composite coating effectively extends the shelf life and maintains the quality of postharvest *T. fuciformis* by modulating ROS metabolism and preserving membrane lipid homeostasis. This study provides a theoretical basis and a practical approach for the quality control of fresh *T. fuciformis*.

## 1. Introduction

*Tremella fuciformis*, commonly known as snow fungus, is an edible and medicinal fungus highly valued for its significant nutritional and health-promoting properties [1]. It is a rich source of bioactive compounds, particularly polysaccharides, as well as proteins, vitamins, minerals, and a diverse range of essential amino acids [2,3]. China is the primary cultivation region for *T. fuciformis*, accounting for the vast majority of global production. For instance, Gutian County, a major production hub, yielded 396,000 t in 2023, representing over 90% of China’s national output. Fresh *T. fuciformis* is typically distributed through cold chain logistics to supermarkets or online platforms for direct consumer access. While this model aims to maintain freshness and brand value, the inherent perishability of the mushroom presents significant challenges during postharvest handling and storage. The promotion of *T. fuciformis* has not only contributed to the prosperity of the market and the fulfillment of consumer demand but has also facilitated an increase in production and income for farmers, thereby positively impacting the agricultural sector and the well-being of farmers. Fresh *T. fuciformis* fruiting bodies are generally white or light yellow. However, improper or prolonged storage leads to quality deterioration, including discoloration to yellowish-brown or dark yellow. Freshly harvested *T. fuciformis* has a high moisture content (around 90%) [4], making it highly susceptible to microbial spoilage, physiological disorders, and metabolic imbalances, all of which contribute to rapid quality decline and reduced shelf life. Under ambient conditions, fresh *T. fuciformis* can rapidly develop undesirable characteristics such as surface browning, hyphal autolysis, and mucus exudation, notably *Burkholderia gladioli* pv. *cocovenenans*. This bacterium can produce Bongkrekic acid, a potent and heat-stable toxin, especially under improper storage conditions such as elevated temperatures and extended durations [5].

A range of preservation techniques are currently employed for edible mushrooms, including physical methods such as irradiation [6], modified atmosphere packaging [7], and cold plasma [8,9,10], as well as chemical treatments like chlorine dioxide [11]. However, these conventional approaches are often associated with significant limitations. Physical methods can be energy-intensive and require specialized equipment, while some chemical preservatives raise environmental and health concerns. Consequently, there is a clear demand for safer, more sustainable, and cost-effective preservation strategies. Edible coatings derived from natural biomaterials represent a particularly promising alternative [12].

Chitosan (CTS), a natural polysaccharide derived from the deacetylation of chitin found in crustacean exoskeletons, is a widely studied biomaterial for food preservation [13]. Owing to its excellent biocompatibility, biodegradability, and low toxicity, CTS has found applications across the medical, food, and agricultural sectors [14]. For postharvest applications, its inherent antimicrobial activity and film-forming properties make it an attractive material for developing edible coatings. These functional attributes can be further enhanced through combination with other active substances, establishing CTS as a promising platform for extending the shelf life of perishable foods [15].

A particularly effective antimicrobial agent for such applications is ε-Poly-L-lysine (ε-PL), a natural homo-poly-amino acid of L-lysine [16]. The potent, broad-spectrum antimicrobial activity of ε-PL stems from its cationic nature, which disrupts the integrity of microbial cell membranes through electrostatic interactions. Recognized as a safe food preservative in many countries due to its excellent biodegradability, low toxicity, and GRAS (Generally Recognized as Safe) status, ε-PL is widely utilized in food preservation [17].

While both CTS and ε-PL exhibit individual preservative effects, their combination in a composite coating is expected to offer enhanced protective advantages for food preservation, potentially enhancing antimicrobial efficacy and improving film properties. Therefore, this study aimed to investigate the effects of an ε-PL/CTS composite coating on the postharvest quality, reactive oxygen species (ROS) metabolism, and membrane lipid metabolism of fresh *T. fuciformis* during storage. The aim was to study the preservation mechanism of this composite coating film on fresh *T. fuciformis* during the storage period. The findings are expected to elucidate the preservation mechanisms of the ε-PL/CTS composite coating on *T. fuciformis*, providing a theoretical basis and practical insights for extending the shelf life of *T. fuciformis* and potentially other edible fungi.

## 2. Results

### 2.1. Effect of ε-PL + CTS Composite Coating on the Quality and Physicochemical Characteristics of T. fuciformis After Harvesting

#### 2.1.1. Weight Loss Rate

Weight loss is a primary indicator of postharvest quality, directly reflecting water loss through transpiration and the consumption of substrates via respiration, both of which contribute to senescence.

The weight loss rate of fresh *T. fuciformis* increased progressively in both the control and ε-PL + CTS-treated groups throughout the 5-day storage period (Figure 1). However, the ε-PL + CTS composite coating significantly (*p* < 0.01 on days 2–4; *p* < 0.05 on day 5) reduced the weight loss rate compared to the control group during the entire storage period, except on day 1 where the difference was not statistically significant. By day 5, the weight loss rate of *T. fuciformis* in the ε-PL + CTS group was 77.75% of that in the control group and was significantly lower (*p* < 0.05). This reduction in weight loss suggests that the ε-PL + CTS coating formed an effective barrier against water transpiration.

#### 2.1.2. Hunter Whiteness

Visual appearance, particularly whiteness, is a critical attribute for the consumer acceptability of *T. fuciformis*. Hunter whiteness provides a comprehensive index to quantify changes in the overall whiteness and color deterioration during storage.

The Hunter whiteness of *T. fuciformis* decreased in both groups during storage, indicating a loss of whiteness (Figure 2). On day 1, Hunter whiteness values for the ε-PL + CTS and control groups were 47.53 and 43.52, respectively, representing a 17.03% and 24.02% reduction from their initial values. Throughout the storage period (days 1–5), *T. fuciformis* treated with ε-PL + CTS consistently maintained higher Hunter whiteness values compared to the control group. Specifically, the ε-PL + CTS group showed significantly higher Hunter whiteness on day 2 and day 5 (*p* < 0.01), and on day 3 and day 4 (*p* < 0.05), compared to the control group. These results suggest that the ε-PL + CTS composite coating effectively preserved the whiteness and visual appearance of *T. fuciformis*.

#### 2.1.3. Browning (*b** Value)

The *b** value, an indicator of yellowness and browning, increased in both treatment groups throughout the storage period (Figure 3). However, the ε-PL + CTS-treated group consistently exhibited significantly lower *b** values compared to the control group from day 2 onwards, indicating less browning. For example, on day 2 and day 3, the *b** values in the ε-PL + CTS group were 5.31 and 6.92, which were 77.18% and 76.72% of the values in the control group, respectively. Statistically, the *b** values in the ε-PL + CTS group were significantly lower than the control group on day 2 and day 4 (*p* < 0.05) and highly significantly lower on day 3 and day 5 (*p* < 0.01).

#### 2.1.4. Changes in Soluble Sugars, Soluble Solids (TSSs), and Soluble Protein Content

Soluble sugars, total soluble solids (TSSs), and soluble proteins are not only key nutritional components but also serve as primary respiratory substrates. Their retention is therefore crucial for maintaining the quality, flavor, and physiological vitality of postharvest *T. fuciformis*.

The soluble sugar content of *T. fuciformis* declined in both ε-PL + CTS and control groups during storage (Figure 4A). By day 5, soluble sugar content decreased to 0.88% and 0.68% in the ε-PL + CTS and control groups, respectively. Throughout the storage period, the ε-PL + CTS group maintained a higher soluble sugar content. For instance, on days 3, 4, and 5, the soluble sugar content in the ε-PL + CTS group was 1.21, 1.29, and 1.31 times higher, respectively, than in the control group. This difference was highly significant (*p* < 0.01) from day 3 to day 5.

TSS content in *T. fuciformis* showed a similar decreasing trend in both groups (Figure 4B). No significant difference in TSS content was observed between the two groups on day 1. On day 2, TSS content decreased to 10.71% in the ε-PL + CTS group and 9.75% in the control group. From day 3 to day 5, while both groups showed a rapid decrease, the ε-PL + CTS group consistently maintained higher TSS levels. By day 5, TSS content in the ε-PL + CTS and control groups had decreased to 7.44% and 6.58%, respectively, representing 65.10% and 57.34% of their initial values on day 0. The TSS content in the ε-PL + CTS group was significantly (*p* < 0.05) higher than that of the control group on day 3.

Soluble protein content in *T. fuciformis* gradually declined during the first 2 days of storage, followed by an accelerated decrease from day 3 to day 5 in both groups (Figure 4C). On day 1, no significant difference in soluble protein content was observed between the treatments. However, from day 3 to day 5, the ε-PL + CTS group maintained higher soluble protein levels. Specifically, on days 3, 4, and 5, soluble protein contents in the ε-PL + CTS group were 5.10, 4.77, and 4.15 mg/g, respectively, which were 1.11, 1.16, and 1.18 times higher than those in the control group. This difference was highly significant (*p* < 0.01) from day 3 to day 5.

These results demonstrate that the ε-PL + CTS composite coating effectively retarded the decline of soluble sugars, TSSs, and soluble proteins in *T. fuciformis* during postharvest storage.

#### 2.1.5. Moisture Migration

The state and distribution of water within *T. fuciformis* tissues are closely linked to its texture, metabolic activity, and shelf life. Low-field nuclear magnetic resonance (LF-NMR) was employed to non-destructively analyze moisture migration and changes in water status among different cellular compartments.

##### Effect of ε-PL + CTS Composite Coating on the Lateral Relaxation Behavior of *T. fuciformis* Hydrogen Protons

A random sample of *T. fuciformis* from the control group and the ε-PL + CTS group, stored under specific conditions (25 °C and relative humidity 80%), was collected on a daily basis. The auricular portion of the same part of *T. fuciformis* was weighed, and the weight was recorded for subsequent peak area normalization calculation. LF-NMR was performed to collect the spectral information and analyze the moisture distribution, phase change and migration pattern of *T. fuciformis* after harvesting and during storage.

The T_2_ relaxation time of fresh *T. fuciformis* showed three hydrogen proton constituent peaks in the range of 0.01~10,000 ms, which were T_21_ (0.01~10 ms), T_22_ (10~100 ms), and T_23_ (100~1000 ms). The shortest relaxation time of T_21_ indicated cell wall-bound water; the second shortest relaxation time of T_22_ indicated cytoplasmic non-flowing water; and the longest relaxation time of T_23_ was mainly the non-flowing water in the tissues of *T. fuciformis*, which had the strongest mobility, indicating that it was the most vulnerable to bacterial utilization, and it was the main factor influencing the storage stability. The peak area of the NMR T_2_ relaxation spectrum was proportional to the number of hydrogen protons in the sample, and the smaller the T_2_ relaxation time was, the greater the constraints on the hydrogen protons, the lower the degree of freedom of the water, the more difficult to remove, and the more to the left of the peak position; and vice versa.

As demonstrated in Figure 5, three distinctive peaks were observed on day 0 of the storage period. The characteristic peak of non-flowing water was the most dominant. The water distribution and migration of fresh *T. fuciformis* from different treatments exhibited analogous trends, albeit with divergent amplitudes. With the passage of time, the signal amplitudes of the three different relaxation peaks decreased and shifted to the left, indicating that the water in *T. fuciformis* moved in the direction of low degree of freedom during the storage and preservation process. Throughout the storage process, the peak area of non-flowing water consistently exhibited the highest percentage, and its change was also the most significant. The signal amplitude of non-flowing water in *T. fuciformis* from both groups of treatments continued to decrease, and T_22_ gradually decreased with the extension of storage time, indicating that the water in this fraction was subjected to an increased binding force and a decreased degree of freedom. The amplitude of the non-flowing water signal of *T. fuciformis* in the ε-PL + CTS group exhibited a more gradual decrease compared with that observed in the control group. The relative content of free water in both groups of *T. fuciformis* gradually increased, but the increase in the ε-PL + CTS group was significantly slower than that in the control group. The cell wall-bound water content exhibited fluctuations during the storage period. However, the 3D water migration waterfall plots in Figure 5 revealed that the cell wall-bound water content of *T. fuciformis* in both groups attained its peak on the 3rd day of storage. The alterations in non-flowing water and free water may be associated with the physiological metabolism and senescence of fresh *T. fuciformis* during postharvest storage. In the course of the storage process, the respiratory metabolism of *T. fuciformis* will also yield a negligible amount of water. This water is continually transformed, thereby enhancing the mobility of *T. fuciformis* tissues and facilitating the interaction of various enzymes in the tissue body and different substrates. This results in vigorous physiological and metabolic activities of the body, which in turn leads to the substrates of *T. fuciformis* constantly losing water and softening. This, in turn, affects the distribution of water in fresh *T. fuciformis* and the phase change and migration patterns.

##### Effect of ε-PL + CTS Composite Coating on *T. fuciformis* Relaxation Time

The three fractions (T_21_, T_22_, and T_23_) of fresh *T. fuciformis* from different treatment groups exhibited a continuous pattern throughout the storage period, with T_2_ demonstrating a general downward trend (Table 1). The fluctuation and decrease in the relaxation time of cell wall water during the storage period was also observed. On the 1st day of storage, a significant decrease (*p* < 0.05) of 49.02% and 55.54% was noted in T_21_ of the ε-PL + CTS group and the control group of *T. fuciformis*, respectively. The T_21_ of both treatment groups of *T. fuciformis* was reduced to the lowest level on the 3rd day of storage, indicating that the mobility of the cell wall water was reduced to a minimum at this time. T_22_ of the two groups demonstrated a decreasing trend during storage, which may be attributable to the substantial dissipation of non-flowing water in the fresh fungus tissues. The T_22_ of the ε-PL + CTS group decreased to 57.960 ms on day 1, while the control group’s T_22_ was only 41.604 ms at this time. Both the ε-PL + CTS group and the control group exhibited a continued decrease in T_23_, reaching 51.25% and 30.43% of the day of harvest and on day 5 of storage, respectively. Consequently, it can be concluded that the ε-PL + CTS composite coating treatment is capable of effectively inhibiting the decline of the mobility of T_21_, T_22_, and T_23_ in *T. fuciformis* during the storage process.

##### Effect of ε-PL + CTS Composite Coating on Unit Peak Area of *T. fuciformis*

The area under the peak of the relaxation spectra is directly proportional to the relative content of water in each component. As demonstrated in Figure 6, the total peak area of both groups of samples exhibited a gradual decrease with increasing storage time. On the 5th day of storage, the total peak area of *T. fuciformis* samples in the ε-PL + CTS group and the control group (A_2_ = A_21_ + A_22_ + A_23_) decreased to 75.30% and 69.09% of the original values, respectively.

The A_21_ content in both the ε-PL + CTS and control groups exhibited a fluctuating upward trend, reaching maximum values of 67.01 and 79.58 on the 3rd day of storage, respectively, and then gradually decreasing in the later stage of storage. This may be due to the decomposition of clustered macromolecules (polysaccharides, proteins) that were tightly bound to cell wall-bound water, resulting in the release of this water from binding and increased mobility; it was transformed into non-flowing water and then into free water. It may also be due to the physiological and metabolic activities of fresh *T. fuciformis*, such as respiration, which led to the vaporization of surface water and the diffusion of free water to the surface to be removed, and the continuous loss of water from the tissues, which led to the gradual reduction in cell wall-bound water as well. The A_22_ content of *T. fuciformis* in the ε-PL + CTS group exhibited a greater magnitude than that of the control group throughout the storage period. The A_23_ content exhibited a marginal increase during the initial 2 days of storage, subsequently reaching a rapid escalation in the 3–5 days following storage. This phenomenon may be attributed to the intensification of cell damage and the augmentation of cell membrane permeability over time, resulting in alterations to the water status. Specifically, non-flowing water underwent a transition into free water. In the late storage period, the inner part of *T. fuciformis* fruiting bodies was decayed and deteriorated by microorganisms and various enzymes, which gradually transformed the bound water and non-flowing water with a high degree of constraint into free water with the lowest degree of constraint, resulting in the gradual increase of A_23_ content. During the entire storage period, the increase in A_23_ content of *T. fuciformis* in the ε-PL + CTS group was less than that in the control group. This indicates that the ε-PL + CTS composite coating treatment was able to inhibit the degree of cellular degradation and maintain a more complete cell membrane structure.

In conclusion, the ε-PL + CTS composite coating treatment effectively retarded the rate of phase state change in each component of fresh *T. fuciformis* during storage, thus inhibiting water migration and water loss of fresh *T. fuciformis*.

#### 2.1.6. Bongkrekic Acid

Fresh *T. fuciformis* is susceptible to spoilage and potential *Bongkrekic acid* production under certain conditions. In this study, despite observing some mold growth in the later stages of storage in both groups, B. acid was not detected (ND) in any samples from either the control or the ε-PL + CTS-treated groups throughout the 5-day storage period at 25 °C (Table 2). The absence of *B. acid* may be attributed to the storage duration being insufficient for its production under the experimental conditions.

### 2.2. Effect of ε-PL + CTS Composite Coating on the Metabolism of Reactive Oxygen Species of T. fuciformis After Harvesting

#### 2.2.1. Superoxide Anion (O_2_^−^) Resistance Capacity

Superoxide anion (O_2_^−^) is one of the primary reactive oxygen species (ROS) produced during senescence, capable of initiating oxidative chain reactions. Therefore, the capacity to resist or scavenge O_2_^−^ is a crucial measure of the *T. fuciformis* ‘s ability to withstand oxidative stress.

The O_2_^−^ resistance of *T. fuciformis* in the control group, measured as U/g protein, initially increased to a maximum of 94.27 between day 1 and day 3 and then decreased to 73.62 by day 5 (Figure 7). Conversely, the ε-PL + CTS-treated group exhibited a markedly enhanced O_2_^−^ resistance, which increased rapidly and continuously from day 0 to day 4, reaching a peak of 127.80 U/g protein on day 4. This peak value was 1.48 times higher than the maximum resistance observed in the control group. Although the resistance in the ε-PL + CTS group decreased to 112.79 U/g protein on day 5, it remained significantly higher (*p* < 0.01) than that of the control group from day 3 to day 5. These results indicate that the ε-PL + CTS composite coating treatment was effective in maintaining a higher superoxide anion resistance in *T. fuciformis*.

#### 2.2.2. Malondialdehyde (MDA) Content

Malondialdehyde (MDA) is a major end-product of membrane lipid peroxidation. Its accumulation serves as a widely used biomarker for the level of oxidative damage to cell membranes caused by ROS.

During postharvest storage, the MDA content increased in both treatment groups (Figure 8). However, the ε-PL + CTS composite coating effectively decelerated the accumulation of MDA in *T. fuciformis*, maintaining it at a lower level compared to the control group. Specifically, on days 2, 3, 4, and 5 of storage, the MDA content in the ε-PL + CTS group was 78.71%, 69.67%, 81.93%, and 81.49% of that in the control group, respectively. The MDA content in the ε-PL + CTS group was significantly lower (*p* < 0.05) than that of the control group on day 2, and highly significantly lower (*p* < 0.01) from day 3 to day 5. These results indicated that the ε-PL + CTS composite coating treatment effectively hindered the escalation of MDA content in *T. fuciformis* postharvest.

#### 2.2.3. Reactive Oxygen Scavenging Enzyme (SOD, CAT, APX) Activity

The enzymatic antioxidant system, including superoxide dismutase (SOD), catalase (CAT), and ascorbate peroxidase (APX), forms the first line of defense against ROS. The activities of these enzymes reflect the cell’s capacity to detoxify harmful oxygen radicals and maintain redox homeostasis.

As shown in Figure 9A, SOD activity in *T. fuciformis* increased rapidly from day 0 to day 1 in both treatments. In the ε-PL + CTS group, SOD activity continued to rise until day 2, reaching a higher level than the control group, where the increase was slower after day 1. Throughout the storage period, SOD activity in the ε-PL + CTS group was consistently higher than in the control group. For example, on day 5, SOD activity in the ε-PL + CTS group was 1.09 times higher than that of the control group. Statistically, SOD activity in the ε-PL + CTS group was significantly higher (*p* < 0.05) than the control group on days 2–3, and highly significantly higher (*p* < 0.01) on days 4–5.

CAT activity in both groups increased initially and then decreased during storage (Figure 9B). The ε-PL + CTS group’s CAT activity peaked at 27.25 U/mg protein on day 1, while the control group’s CAT activity peaked at 24.51 U/mg protein on day 2. CAT activity in the ε-PL + CTS group was consistently higher than in the control group. Notably, on day 1, CAT activity in the ε-PL + CTS group was 1.39 times higher than that of the control group and the difference was highly significant (*p* < 0.01).

APX activity declined with the extension of storage time in both treatments (Figure 9C). On day 1, APX activity in the control group decreased rapidly to 0.24 U/mg protein, while in the ε-PL + CTS group it was 0.40 U/mg protein, 1.67 times higher. APX activity in the ε-PL + CTS group remained higher than that of the control group from day 1 to day 5. This difference was significant (*p* < 0.05) on days 1, 3, and 5, and highly significant (*p* < 0.01) on day 4.

In summary, the ε-PL + CTS treatment group maintained higher levels of SOD, CAT, and APX enzyme activities in postharvest *T. fuciformis*.

#### 2.2.4. Reactive Oxygen Non-Enzymatic Scavengers (AsA and GSH) Content

Non-enzymatic antioxidants, such as ascorbic acid (AsA) and reduced glutathione (GSH), play a vital role in directly scavenging ROS and regenerating other antioxidants. Their content is indicative of the overall non-enzymatic antioxidant potential of the tissue.

The AsA content in the control group of *T. fuciformis* decreased to 15.50 mg/100 g by day 2 (a 50.84% reduction from day 0), then slightly increased between days 2 and 4, before rapidly declining to 10.89 mg/100 g by day 5 (Figure 10A). In the ε-PL + CTS group, the initial decrease in AsA content on day 1 was not significant compared to the control’s inhibition. However, AsA content then rapidly increased to a peak of 25.15 mg/100 g on day 4, before decreasing to 19.19 mg/100 g on day 5. The AsA content in the ε-PL + CTS group was significantly higher (*p* < 0.05) than the control group on day 4, and highly significantly higher (*p* < 0.01) on day 5, being 1.76 times greater.

The GSH content in both groups exhibited a transient increase on day 1, followed by a rapid decline from day 2 to day 5 (Figure 10B). The lowest GSH values of 11.17 µmol/g (ε-PL + CTS) and 8.60 µmol/g (control) were recorded between days 2 and 5. Throughout the storage period, the GSH content in the ε-PL + CTS group was consistently higher than in the control group. For instance, on days 3, 4, and 5, GSH content in the ε-PL + CTS group was 1.28, 1.31, and 1.30 times higher, respectively, than in the control group. These increases were highly significant (*p* < 0.01) from day 2 to day 5.

These findings indicated that the ε-PL + CTS composite coating treatment was effective in maintaining higher levels of AsA and GSH, important non-enzymatic antioxidants, in *T. fuciformis* after harvest.

#### 2.2.5. DPPH Radical Scavenging and Reducing Power

The DPPH radical scavenging activity and reducing power are important assays that reflect the total antioxidant capacity of the *T. fuciformis* extract, encompassing the combined effects of all antioxidant compounds, both enzymatic and non-enzymatic.

The DPPH radical scavenging capacity of *T. fuciformis* in the ε-PL + CTS group was consistently higher than that of the control group throughout storage (Figure 11A). In the ε-PL + CTS group, DPPH scavenging capacity increased to a maximum of 142.68 µg TROLOX/g on day 1, decreased rapidly between days 1 and 3, and then showed a slow increase from day 3 to day 5. In contrast, the control group exhibited a rapid decrease in DPPH scavenging capacity from day 0 to day 2, followed by a slower decline. On days 3, 4, and 5, the DPPH scavenging capacity in the ε-PL + CTS group was 1.78, 2.15, and 2.07 times higher, respectively, than in the control group. This difference was significant (*p* < 0.05) on day 1 and highly significant (*p* < 0.01) from day 2 to day 5.

The reducing power of *T. fuciformis* in the control group showed a fluctuating decreasing trend during storage, while the ε-PL + CTS group demonstrated a slowly increasing trend (Figure 11B). On days 4 and 5, the reducing power in the ε-PL + CTS group was 1.56 and 1.72 times higher, respectively, than that of the control group. The reducing power in the ε-PL + CTS group was highly significantly (*p* < 0.01) higher than that of the control group from day 2 to day 5.

These results indicated that the ε-PL + CTS composite coating treatment enhanced the DPPH radical scavenging ability and reducing power of postharvest *T. fuciformis*.

### 2.3. Effect of ε-PL + CTS Composite Coating on Lipid Metabolism of Post-Harvest T. fuciformis Membranes

#### 2.3.1. Cell Membrane Permeability

Cell membrane permeability, measured as relative electrolyte leakage, is a key indicator of membrane integrity. An increase in permeability signifies membrane damage and a loss of cellular compartmentalization, which is a hallmark of senescence.

With the prolongation of storage time, cell membrane permeability of *T. fuciformis* increased in both treatment groups, although the magnitudes differed (Figure 12). In the control group, cell membrane permeability exhibited a rapidly rising trend, increasing to 87.02% by day 5, a 54.73% increase over the storage period. Conversely, the ε-PL + CTS group showed a more gradual increase in cell membrane permeability, reaching 76.81% on day 5. Throughout the entire storage period, the cell membrane permeability of *T. fuciformis* in the ε-PL + CTS group was highly significantly (*p* < 0.01) lower than that of the control group. These results indicated that the ε-PL + CTS composite coating treatment effectively inhibited the increase in cell membrane permeability of postharvest *T. fuciformis*.

#### 2.3.2. Membrane Lipid Degradation-Related Enzyme (Lipase, LOX, Phospholipase PLD) Activities

The stability of cell membranes is closely regulated by the activity of lipid-degrading enzymes. Lipase, lipoxygenase (LOX), and phospholipase D (PLD) can accelerate membrane deterioration by hydrolyzing or oxidizing membrane lipids, making their activities critical markers of membrane stability.

As demonstrated in Figure 13A, lipase activity in *T. fuciformis* exhibited a similar trend in both treatments, increasing to a peak on day 3 and then declining. The ε-PL + CTS group and the control group attained maximum lipase activities of 11.96 and 13.00 U/mg protein, respectively, on day 3, subsequently decreasing to 8.20 and 9.58 U/mg protein by day 5. While the coating’s effect on inhibiting lipase activity was not significant on day 1, by day 5, lipase activity in the ε-PL + CTS group was only 85.62% of that in the control group, and this difference was highly significant (*p* < 0.01).

LOX activity in the control group increased rapidly during the first 3 days of storage, reaching a maximum of 64.59 U/mg protein, followed by a rapid decrease to 38.99 U/mg protein by day 5 (Figure 13B). In contrast, the ε-PL + CTS group maintained a consistently lower level of LOX activity. On days 2, 3, and 4, LOX activity in the ε-PL + CTS group was 70.82%, 71.11%, and 70.12% of that in the control group, respectively. The LOX enzyme activity in the ε-PL + CTS group was significantly lower (*p* < 0.05) than the control group on day 5, and highly significantly lower (*p* < 0.01) from day 2 to day 4.

PLD activity in *T. fuciformis* markedly increased during the initial 3 days of storage, followed by a rapid decline between days 4 and 5 (Figure 13C). Notably, PLD activity in the ε-PL + CTS-treated samples remained consistently lower throughout the storage period. On days 4 and 5, PLD activity in the ε-PL + CTS group was 65.34% and 71.96% of that observed in the control group, respectively. Statistical analysis demonstrated that PLD activity in the ε-PL + CTS group was highly significantly lower (*p* < 0.01) than that of the control group from day 3 to day 5.

These findings indicated that the ε-PL + CTS composite coating treatment effectively inhibited the increase in the activities of membrane lipid degradation-related enzymes (lipase, LOX, and PLD), thereby slowing the rate of membrane lipid degradation in *T. fuciformis* during postharvest storage.

#### 2.3.3. Phospholipid Fraction Content

Phospholipids, such as phosphatidylinositol (PI) and phosphatidylcholine (PC), are the fundamental structural and functional components of cell membranes. The degradation of these phospholipids and the accumulation of their byproducts, like phosphatidic acid (PA), directly reflect the state of membrane integrity and the progression of senescence.

PI content in *T. fuciformis* continuously decreased during storage in both treatments (Figure 14A). No significant difference in PI content was observed between the two treatments on day 1. However, from day 1 to day 5, PI content in the ε-PL + CTS group remained at a higher level than in the control group. Specifically, on days 3, 4, and 5, PI content in the ε-PL + CTS group was 1.61, 2.46, and 2.01 times higher, respectively, than in the control group. The PI content in the ε-PL + CTS group was significantly higher (*p* < 0.05) than the control group on days 2–3, and highly significantly higher (*p* < 0.01) on day 4.

PC content exhibited a rapid decline during the initial 3 days of storage, followed by a more gradual decrease from day 4 to day 5 in both groups (Figure 14B). The PC content in the ε-PL + CTS group remained higher than that of the control group throughout the entire storage period. On day 4, PC content in the ε-PL + CTS group was 2.22 times higher than in the control group, and this difference was significant (*p* < 0.05).

PA content in the control group rapidly increased to 0.085 mg/g between day 0 and day 2, followed by a more gradual rise until day 5 (Figure 14C). In comparison, PA content in the ε-PL + CTS group showed a comparatively gradual increase, reaching a maximum of 0.076 mg/g on day 3, and then fluctuated slightly until day 5. The PA content in the ε-PL + CTS group was significantly lower (*p* < 0.05) than that of the control group on day 3, and highly significantly lower (*p* < 0.01) on days 2 and 5, being only 65.47% and 74.22% of the control group’s values on these respective days.

These results indicated that the ε-PL + CTS composite coating treatment maintained higher PI and PC contents and delayed the increase in PA content in *T. fuciformis* after harvesting.

## 3. Discussion

The ε-PL/CTS composite coating was found to significantly preserve the postharvest quality of *T. fuciformis*. Our findings indicate that this preservative effect is strongly associated with the coating’s ability to modulate two interconnected physiological processes: reactive oxygen species (ROS) metabolism and membrane lipid homeostasis. It is well-established that these two pathways are intricately linked, as oxidative stress induced by excessive ROS can trigger membrane lipid peroxidation, leading to cellular dysfunction and accelerated senescence [18].

Postharvest senescence in edible fungi is often accompanied by an upsurge in respiration and a concomitant overproduction of ROS, such as O_2_^−^ and H_2_O_2_ [18,19]. O_2_^−^ can initiate lipid peroxidation, while malondialdehyde (MDA), a product of lipid peroxidation, reflects the extent of oxidative damage [20]. Effective ROS management is crucial for postharvest quality preservation, with the control of superoxide anion (O_2_^−^) being a critical first step. In this study, the ε-PL/CTS composite coating significantly enhanced the O_2_^−^ resistance capacity of *T. fuciformis* (Figure 7), which in turn correlated with lower MDA accumulation, a key marker of oxidative damage (Figure 8). This enhanced resistance can be attributed to the modulation of the enzymatic antioxidant system. SOD is a crucial intracellular antioxidant enzyme that catalyzes the disproportionation reaction of superoxide anion to generate H_2_O_2_ and O_2_, thereby reducing the accumulation of superoxide anion. APX and CAT further decompose H_2_O_2_ into H_2_O and O_2_, thereby reducing the peroxidative stress and cell membrane damage caused by the accumulation of reactive oxygen species [21]. Our results showed that the ε-PL/CTS coating maintained significantly higher activities of SOD, CAT, and APX in *T. fuciformis* throughout storage (Figure 9A–C). This sustained enzymatic activity likely contributed to more efficient ROS scavenging. The improved enzyme performance in coated samples could be due to several factors: the chitosan-based film may have created a modified internal atmosphere, reducing basal ROS generation by moderating respiration, while the antimicrobial action of ε-PL and chitosan could have lessened microbe-induced oxidative stress. By reducing the overall oxidative load, the coating may have helped preserve these enzymes from inactivation or depletion. These findings are consistent with Ma et al. [22], who reported that decompression treatment enhanced SOD, CAT, and APX activities and inhibited O_2_^−^ and MDA production in postharvest *Pleurotus eryngii*, thereby delaying senescence. Beyond the enzymatic defenses, non-enzymatic antioxidants such as ascorbic acid (AsA) and glutathione (GSH) play vital roles in cellular protection. AsA directly scavenges various ROS, while GSH participates in reducing H_2_O_2_ and lipid peroxides, and acts as a cofactor for antioxidant enzymes like glutathione peroxidase [23]. In the present study, the ε-PL/CTS coating treatment led to significantly higher levels of both AsA and GSH in *T. fuciformis* throughout storage (Figure 10A,B), further bolstering the *T. fuciformis*’s overall antioxidant system. This aligns with research by Wang et al. [24], where a combined 1-MCP and nitric oxide treatment enhanced AsA and GSH contents in stored *Agaricus bisporus*, effectively reducing ROS accumulation and maintaining quality. Furthermore, the overall antioxidant potential, as assessed by DPPH radical scavenging activity and reducing power, was significantly improved in coated *T. fuciformis* (Figure 11A,B). These parameters reflect the collective ability of various antioxidant components to neutralize free radicals and reduce oxidants [25]. Our results are supported by Dokhanieh et al. [26], who found that salicylic acid treatment enhanced DPPH radical scavenging capacity and total antioxidant activity in postharvest *A. bisporus*, which correlated with delayed browning. The enhanced antioxidant status in ε-PL/CTS coated *T. fuciformis* indicates a comprehensive protective effect against oxidative deterioration.

Cell membrane integrity is paramount for maintaining the quality of postharvest edible fungi, and its stability is largely dependent on the homeostasis of membrane lipids [27]. Increased cell membrane permeability, reflecting loss of integrity and leakage of cellular contents, is a common symptom of senescence [28]. Our study demonstrated that the ε-PL/CTS coating significantly retarded the increase in cell membrane permeability in *T. fuciformis*, indicating better preservation of membrane integrity. This preservation of membrane integrity was closely linked to the coating’s ability to inhibit the activities of key membrane lipid-degrading enzymes: lipoxygenase (LOX), lipase, and phospholipase D (PLD). LOX catalyzes the oxidation of polyunsaturated fatty acids, initiating peroxidative damage. Lipase hydrolyzes triacylglycerols, and PLD degrades phospholipids; elevated activities of these enzymes contribute to the structural and functional collapse of cell membranes [27,29]. The suppressed activities of LOX, lipase, and PLD in coated *T. fuciformis* suggest a reduced rate of membrane lipid degradation. This enzymatic inhibition is likely a consequence of the multifaceted protection conferred by the coating, including the aforementioned reduction in ROS-mediated oxidative stress (which enhances the resistance of membranes to peroxidation), which can damage both lipids and these enzymes. Furthermore, the physical barrier provided by the coating might alter the microenvironment at the mushroom surface, indirectly influencing enzyme activity. The maintenance of lower LOX, lipase, and PLD activities contributed to the reduced rate of membrane lipid breakdown, thereby preserving membrane structure and function. These observations align with findings by Ma et al. [29], who showed that nano Ag/TiO_2_ polyethylene-based packaging inhibited PLD, lipase, and LOX activities in *A. bisporus*, leading to improved storage quality. Further supporting the role of the ε-PL/CTS coating in membrane stabilization, our analysis of phospholipid fractions revealed that the coating helped maintain higher levels of phosphatidylcholine (PC) and phosphatidylinositol (PI), which are major structural and functional components of eukaryotic cell membranes. Conversely, the accumulation of phosphatidic acid (PA), a product of phospholipid hydrolysis that can exacerbate membrane damage when present in excess [30], was significantly delayed in coated samples. The preservation of PC and PI, coupled with the attenuated PA accumulation, underscores the coating’s effectiveness in protecting the phospholipid bilayer and enhancing its resistance to degradation. The overall success of chitosan-based composite coatings in retaining the quality of other mushroom species, such as button mushrooms and oyster mushrooms, has also been documented [31,32], lending further credence to our observations.

In conclusion, this study demonstrates that the ε-PL/CTS composite coating is an effective edible film for extending the postharvest shelf life and maintaining the quality of *T. fuciformis*. The protective mechanism appears to involve a combined action of ε-PL and CTS, which creates a favorable microenvironment around the *T. fuciformis*. This leads to a significant reduction in oxidative stress by limiting ROS generation and enhancing both enzymatic and non-enzymatic antioxidant capacities. Consequently, the integrity of cell membranes is better preserved through the inhibition of lipid peroxidation and the suppression of membrane-degrading enzyme activities, ultimately delaying senescence and quality deterioration.

## 4. Materials and Methods

### 4.1. Materials and Reagents

Fresh fruiting bodies of *T. fuciformis* (cultivar Tr21) were harvested from a commercial cultivation base in Gutian County, Fujian Province, China, and mature *T. fuciformis* with intact morphology, uniform size, glossy appearance, no mechanical damage, and free of mold and insect pests were selected for subsequent experimental treatments.

ε-Poly-L-lysine (ε-PL, molecular weight 3–5 kDa) and chitosan (CTS, molecular weight 50–90 kDa, degree of deacetylation 90%) were purchased from Zhejiang Yinuo Biotechnology Co., Ltd. (Lanxi, China). Kits for the determination of superoxide anion (O_2_^−^) generation and inhibition, total superoxide dismutase (T-SOD) activity, ascorbate peroxidase (APX) activity, and 2,2-diphenyl-1-picrylhydrazyl (DPPH) radical scavenging ability were obtained from Nanjing Jiancheng Bioengineering Institute (Nanjing, China). All other chemicals and reagents used were of analytical grade and purchased from Shanghai Macklin Biochemical Technology Co., Ltd. (Shanghai, China).

### 4.2. Instruments and Equipment

High-speed refrigerated centrifuge (GL-21M, Hunan Xiangyi Laboratory Instrument Development Co., Ltd., Changsha, China). Constant temperature water bath (Model DF-101S, Shanghai Lichen Bangxi Instrument Technology Co., Ltd., Shanghai, China). Conductivity meter (Model DDS-11C, Beijing Puci Technology Co., Beijing, China). High-Performance Liquid Chromatography (HPLC) system (LC-2030C, Shimadzu, Kyoto, Japan) equipped with an Evaporative Light Scattering Detector (ELSD-LT II, Shimadzu, Kyoto, Japan). Colorimeter (NS810, Shenzhen Threenh Technology Co., Ltd., Shenzhen, China). Low field nuclear magnetic resonance (LF-NMR) instrument (MesoMR23-060H-I, Suzhou Niumag Analytical Instrument Corporation, Suzhou, China).

### 4.3. Determination of Optimal Coating Concentration

The selection of the coating concentration for this study was based on a series of preliminary experiments. Building on previous findings that a composite ε-PL/CTS coating is more effective than either ε-PL or CTS applied individually, we designed a matrix of 12 different formulations. These formulations combined various concentrations of ε-polylysine (ε-PL: 50, 100, 150, and 200 mg/L) with different concentrations of chitosan (CTS: 1, 5, and 10 g/L). Through this screening process, the formulation of 150 mg/L ε-PL combined with 5 g/L CTS was identified as the optimal concentration. Consequently, this concentration was used for all subsequent in-depth analyses in this study

### 4.4. Formal Trials for This Study

Freshly harvested *T. fuciformis* fruiting bodies were randomly divided into two groups (*n* = 120 per group). The control group (denoted as Control group) was treated with a uniform spray of sterile water, while the treatment group was sprayed with a 150 mg/L ε-PL + 5 g/L CTS composite coating. After sufficient drying, the samples were placed in polyethylene film preservation bags and stored at (25 ± 1) °C and 80% relative humidity. Samples were taken daily and the relevant indexes were measured.

### 4.5. Determination of Physicochemical Properties and Physiological Parameters

#### 4.5.1. Measurement of Weight Loss Rate

For each treatment, three replicates (each replicate containing 3 fruiting bodies) were used for weight loss determination and weighed at day 0 (initial weight) and at each sampling point during storage, and weightlessness was calculated using the following formula:$$Weight\;Loss\;Rate(\%)=\frac{m_{0}-m_{1}}{m_{0}}\times 100\%$$
where m_0_ is the initial weight of the sample (g) and m_1_ is the weight of the sample at a specific storage time (g).

#### 4.5.2. Measurement of Hunter Whiteness

Color was measured using a NS810 colorimeter, and the CIELAB parameters *L**, *a**, and *b** were recorded. Hunter whiteness was then calculated from these values as a comprehensive measure of the sample’s whiteness using the following formula:$$Hunter\;whiteness=100-\sqrt{{\left(100-L^{*}\right)}^{2}+{a^{*}}^{2}+{b^{*}}^{2}}$$
where *L** is the surface light of *T. fuciformis*; *a** is the red-green value; *b** is the yellow-blue value.

#### 4.5.3. Measurement of Browning Condition (*b** Value)

The *b** value, representing yellowness, was used as an indicator of browning and measured as described in Section 4.5.2.

#### 4.5.4. Determination of Soluble Sugars, Soluble Solids, and Soluble Protein Content

##### Determination of Soluble Sugar Content

Soluble sugar content was determined using the anthrone-sulfuric acid method as described by Luo et al. [33] with some modifications. A total of 0.1 g of *T. fuciformis* sample was homogenized with 1.5 mL of distilled water and extracted in a boiling water bath for 10 min. The mixture was centrifuged at 8000 rpm for 10 min at 4 °C. Subsequently, 0.1 mL of the supernatant was taken, and the solution to be measured was obtained by mixing with 0.9 mL of distilled water. A volume of 0.5 mL of the solution to be measured was taken, followed by the addition of 1.5 mL of distilled water, 0.5 mL of anthrone–ethyl acetate reagent, and 5 mL of concentrated H_2_SO_4_. These were mixed thoroughly and immediately transferred to a boiling water bath for 1 min, after which they were cooled down to room temperature. The absorbance was determined at 630 nm and calculated using the following formula:$$Soluble\;Sugars(\%)=\frac{m\times V\times N}{V_{1}\times m_{1}\times 10^{6}}\times 100\%$$
where C is the concentration of sugar from the standard curve (μg/mL), V is the total volume of the sample extract (mL), D is the dilution factor, and W is the fresh weight of the sample (g).

##### Determination of Soluble Solids

A total of 10 g of *T. fuciformis* sample was homogenized with 30 mL of distilled water. The mixture was subjected to thorough grinding and centrifugation at 5000 rpm for 10 min at 4 °C. The supernatant was then obtained and measured using a handheld refractometer [34]. The formula was calculated as follows:$$soluble\;solids(\%)=\frac{m_{0}+m_{1}}{m_{1}}\times P$$
where P is the soluble solid content of the sample liquid (%); m_0_ is the mass of the sample (g); m_1_ is the mass of distilled water added to the sample (g).

##### Determination of Soluble Protein Content

Soluble protein content was determined by the Coomassie Brilliant Blue G-250 dye-binding method according to Ma et al. [35] with modifications, using bovine serum albumin (BSA) as the standard. A total of 2 g of *T. fuciformis* sample was homogenized in 40 mL of distilled water and homogenized in an ice bath. The mixture was then centrifuged at 12,000 rpm for 20 min at 4 °C. A volume of 1 mL of the supernatant was pipetted and combined with 5 mL of Coomassie brilliant blue G-250 solution. The contents were mixed thoroughly and allowed to stand for a period of 2 min. The absorbance was then measured at 595 nm, and the protein content was calculated using the standard curve.

#### 4.5.5. Determination of Water Status by LF-NMR

The transverse relaxation time (T_2_) spectra were acquired following the initial determination of the center frequency and pulse width through the free induction decay (FID) sequence. The sample was then placed within the magnet box to collect the relaxation signal in accordance with the established protocols. The probe option NMI20-040H-I-40 mm was selected, and the Carr–Purcell–Meiboom–Gill (CPMG) sequence was used with the following parameters: proton resonance frequency (SF) = 20 MHz; RF delay, 0.1 ms; frequency bias, 628,744.24 hz; analogue gain, 20 dB; 90-degree pulse width, 6.6 µs; digital gain, 3; number of sampling points, 540,014; data radius, 1; pre-release gear, 1; waiting time, 4000 ms; cumulative number of times, 3180 degrees; pulse width, 11.6 us; echo time, 0.6 ms; number of echoes, 9000; and peak offset, 0 ms. Once the measurement was complete, all peak points were selected, the number of data points was specified as 200, the basal sampling data was set to 0, the filtering gear was set to 0, and the minimum value of relaxation time was defined as 0.01 ms, with a maximum value of 1. The relaxation time had a total of 200 points, with a display value limit of 0.1 a.u. The SIRT inversion method was selected, with a multicomponent inversion number of 100,000.

#### 4.5.6. Determination of Bongkrekic Acid

Bongkrekic acid (BA) content was determined according to the National Food Safety Standard of China (GB 5009.189-2023; Determination of Bongkrekic acid in foods. National Health Commission of the People’s Republic of China & State Administration for Market Regulation: Beijing, China, 2023), using Method 1 (Liquid Chromatography). The limit of detection (LOD) was 0.015 mg/kg.

#### 4.5.7. Determination of Superoxide Anion (O_2_^−^) Resistance Capacity

The capacity of *T. fuciformis* extracts to counteract O_2_^−^ was assessed using a commercial kit, and the anti-O_2_^−^ capacity during storage was calculated in accordance with the instructions provided with the kit.

#### 4.5.8. Determination of Malondialdehyde (MDA) Content

MDA content was determined using the thiobarbituric acid (TBA) method as described by Huang et al. [36] with slight modifications. A total of 5 g of *T. fuciformis* sample was weighed and added to 30 mL of trichloroacetic acid (TCA) solution. The mixture was then ground in an ice bath to ensure homogenization and subsequently centrifuged at 10,000 rpm for 20 min at 4 °C. A total of 2 mL of the supernatant was taken, and 2 mL of 0.67% thiobarbituric acid (TBA) was added. The solution was then stirred uniformly and boiled for 20 min, after which it was cooled and the absorbance at 600 nm, 532 nm, and 450 nm was determined.

#### 4.5.9. Determination of Reactive Oxygen Scavenging Enzymes (SOD, CAT, APX)

##### Determination of Superoxide Dismutase (SOD) Activity

The test was conducted using a total superoxide dismutase (T-SOD) test kit, and the activity of SOD was calculated in accordance with the instructions provided with the kit.

##### Determination of Catalase (CAT) Activity

In accordance with the methodology proposed by He et al. [37], 1 g of *T. fuciformis* and 3 mL of a pH 7.0 phosphate buffer solution (PBS) were combined and homogenized in an ice bath. The resulting mixture was then subjected to centrifugation for at 8000 rpm for 10 min at 4 °C, after which the supernatant was collected and designated as the enzyme solution. A volume of 0.1 mL of the enzyme solution was taken and combined with 1.4 mL of 20 mmol/L H_2_O_2_; the resulting mixture was incubated at 30 °C for 10 min. The inactivated enzyme solution was used as a reference.

##### Determination of Ascorbate Peroxidase (APX) Activity

The APX assay kit was used to perform the requisite tests, with the resulting APX activity calculated in accordance with the instructions provided with the kit.

#### 4.5.10. Determination of Reactive Oxygen Non-Enzymatic Scavenging Substances (AsA, GSH)

##### Determination of Reduced Ascorbic Acid (AsA) Content

Reduced ascorbic acid (AsA) content was determined using the method described by Wang et al. [38] with modifications. A total of 5 g of *T. fuciformis* sample was homogenized with 35 mL of pre-cooled 5% (*w*/*v*) TCA solution. The mixture was then ground and crushed under an ice bath, and the centrifugation parameters were set at 12,000 rpm for 10 min at 4 °C. A volume of 3 mL of the resulting supernatant was taken and combined with 3 mL of 5% TCA and 3 mL of anhydrous ethanol. After vortexing and thorough mixing, 1.5 mL of 0.4% H_3_PO_4_–ethanol, 3 mL of 0.5% ophiophenanthroline–ethanol, and 1.5 mL of 0.03% FeCl_3_–ethanol were added. A solution of 0.5% o-phenanthroline–ethanol (1.5 mL) and 0.03% FeCl_3_–ethanol (1.5 mL) was vortexed and mixed again and then incubated at 30 °C for 90 min. The OD values were subsequently measured at 534 nm. A standard curve was prepared using L-ascorbic acid. AsA content is expressed in mg/100 g.

##### Determination of Reduced Glutathione (GSH) Content

A volume of 5 g of *T. fuciformis* was added to 35 mL of a pre-cooled 5% (*w*/*v*) TCA solution containing 5 mM EDTA-Na_2_, homogenized at high speed in an ice bath, and then centrifuged at 12,000 rpm for 10 min at 4 °C.

Two test tubes were prepared, with 1 mL of supernatant and 1 mL of 0.1 mol/L PBS (pH 7.7) added to each. One tube was then supplemented with 0.5 mL of 4 mmol/L dithionitrobenzoic acid (DTNB) solution, while the other was left as a control. A volume of 0.1 mol/L PBS (pH 6.8) was added to the other tube, and the two tubes were maintained at 25 °C for 10 min. The OD value was then measured at 412 nm, and a standard curve was constructed with the GSH standard, with the results expressed in mmol/g [39].

#### 4.5.11. Determination of 2,2-Diphenyl-1-Trinitrohydrazine (DPPH) Radical Scavenging Activity and Reducing Power

DPPH radical scavenging activity was assayed using a commercial kit according to the manufacturer’s instructions. Briefly, the assay measures the ability of the sample extract to scavenge the stable DPPH radical, monitored by the decrease in absorbance at 517 nm.

Reducing power was determined by the potassium ferricyanide method as described by Yin et al. [40]. A total of 1 g of *T. fuciformis* sample was homogenized with 3 mL of 80% (*v*/*v*) ethanol and centrifuged at 8000 rpm for 10 min at 4 °C. The supernatant was used for the assay. A volume of 0.2 mL of the supernatant was combined with 2.5 mL of PBS (pH 6.6) and 2.5 mL of 1% potassium ferricyanide, vortexed, and mixed thoroughly. Subsequently, 2.5 mL of 10% TCA was added. The mixture was then cooled to 4 °C and centrifuged at 6000 rpm for 10 min at 4 °C. A total of 1 mL of the resulting supernatant was combined with 1 mL of distilled water and 1 mL of 0.1% FeCl_3_ and was vortexed and mixed well. Higher absorbance at 700 nm indicates greater reducing power.

#### 4.5.12. Determination of Cell Membrane Permeability

The method described by Yang et al. [41] was modified slightly. A volume of 5 g of *T. fuciformis* sample was immersed in 20 mL of deionized water and incubated at 25 °C for 3 h with occasional shaking. After thorough mixing, the conductivity value was determined by a conductivity meter and recorded as C_1_. Subsequently, the sample was boiled for 30 min to release all electrolytes, then cooled to room temperature, and the volume was made up to 20 mL with deionized water. The conductivity value was then determined by the same method and recorded as C_2_. The permeability of cell membranes can be characterized in terms of the relative osmotic permeability.$$Cell\;membrane\;permeability(\%)=\frac{C_{1}}{C_{2}}\times 100\%$$

#### 4.5.13. Determination of Membrane Lipid Degrading Enzyme Activities

##### Determination of Lipase Activity

In accordance with the methodology proposed by Chen et al. [42], 1 g of *T. fuciformis* sample was homogenized in 3 mL of 0.2 M phosphate buffer (pH 7.8) containing 0.05 M β-mercaptoethanol. The homogenate was centrifuged at 8000 rpm for 10 min at 4 °C. The supernatant was used as the enzyme extract. Subsequently, 1 mL of the enzyme extract was mixed with 2.3 mL of 0.2 M PBS (pH 7.8) and pre-incubated at 30 °C for 5 min. Then, 0.5 mL of 0.5% (*w*/*v*) α-naphthyl acetate was added as substrate, vortexed, and incubated at 30 °C for 30 min. The reaction was stopped by adding 0.2 mL of 0.15% (*w*/*v*) Fast Blue B salt solution (containing 6% SDS). The absorbance was measured at 520 nm.

##### Determination of Lipoxygenase (LOX) Activity

Modifications were made with reference to the methods proposed by Hao [43] and Zhou [44]. A volume of 1 g of *T. fuciformis* sample was homogenized in 3 mL of pre-cooled 0.1 M phosphate buffer (pH 6.8) containing 1% (*v*/*v*) Triton X-100 and 4% (*w*/*v*) PVPP. The solution was then subjected to centrifugation at 8000 rpm for 10 min at 4 °C. A solution of sodium linoleate (50 μL) was combined with 0.2 mL of the supernatant, and the mixture was incubated for 5 min at a constant temperature of 30 °C. Subsequently, 4.5 mL of 70% ethanol was added to the solution, and the OD 234 nm was measured to determine the OD value, with an inactivated enzyme solution serving as the reference. A change of 0.01 in the absorbance value per minute was taken as the unit of enzyme activity (U), with the results expressed as U/mg protein.

##### Phospholipase D (PLD) Activity Assay

The method described by Yu et al. [45] was modified as follows: 1 g of *T. fuciformis* sample was homogenized in 4 mL of pre-cooled 0.1 M acetate buffer (pH 5.6) containing 5 mM DTT and 1 mM CaCl_2_. The homogenate was centrifuged at 8000 rpm for 10 min at 4 °C. The supernatant was used as the enzyme extract. Subsequently, 1 mL of the supernatant was transferred to a new tube, 1 mL of a 0.4 mg/mL lecithin solution was added, and the mixture was incubated at 28 °C for 1 h. Finally, the lower layer was washed with petroleum ether three times, after which the lower layer was collected. A further 1 mL of 1% Reisner’s salt was added, and the mixture was subjected to a second centrifugation at 6000 rpm for 10 min. Following this, 2 mL of acetone was added to the precipitate, which was then dissolved, and the OD value was measured at 520 nm. The inactivated enzyme solution was employed as a reference point. A change of 0.01 in absorbance per minute was taken as one unit of enzyme activity (U), with the results expressed as U/mg protein.

#### 4.5.14. Determination of Phospholipid Composition

In accordance with the methodologies proposed by Li et al. [46] and Niu et al. [47], lipids were extracted from 1 g of *T. fuciformis* sample with 3 mL of chloroform/methanol (2:1, *v*/*v*) using ultrasonication in an ice bath for 1 h. After centrifugation (10,000 rpm, 10 min, 4 °C), the lower chloroform phase containing lipids was collected. The lower chloroform phase was removed, 1 mL of acetone was added, and the solution was vortexed and mixed for two minutes. This step was repeated three times before the liquid was blown dry using a nitrogen blower. The lipid residue was redissolved in 1 mL of chloroform/methanol (9:1, *v*/*v*) for HPLC analysis.

The phospholipid fractions were determined by the HPLC-ELSD method. The column model was an Inersil SIL 100A silica gel column (4.6 × 250 mm), the column temperature was 30 °C, and the injection volume was 10 µL. The wavelength was 254 nm, the flow rate was 0.8 mL/min, and the mobile phases were n-hexane–isopropanol–methanol–1% acetic acid (4:9:5:2, *v*/*v*/*v*/*v*/*v*). The retention time of the standard samples of phosphatidylcholine (PC), phosphatidylinositol (PI), and phosphatidic acid (PA) was employed for qualitative analysis, while the standard curves of PI, PA, and PC specimens with a concentration range of 0–2 mg/mL were utilized for quantitative analysis. The results were expressed as mg/g.

#### 4.5.15. Statistical Analysis

Data were analyzed using SPSS software (Version 21.0, IBM Corp., Armonk, NY, USA) and figures were plotted using GraphPad Prism (Version 8.0.2, GraphPad Software, San Diego, CA, USA) and Origin (Version 2017, OriginLab Corporation, Northampton, MA, USA). The differences between the means were compared using the independent samples *t*-test. All experiments were performed in triplicate, and data are presented as mean ± standard deviation (SD). Statistical significance was determined using the a*t*-test. Differences were considered statistically significant at *p* < 0.05 and highly significant at *p* < 0.01.

## 5. Conclusions

Fresh *T. fuciformis* is highly perishable at room temperature, which limits its shelf life. This study demonstrated that an ε-PL and CTS composite coating effectively preserved the postharvest quality of *T. fuciformis*. The coating significantly reduced weight loss, maintained better visual appearance (color and whiteness), and retarded the degradation of key nutritional components, including soluble solids, soluble proteins, and soluble sugars. Moreover, the treatment effectively controlled water status by slowing water migration and loss from *T. fuciformis* tissues.

The preservative effect of the ε-PL/CTS composite coating was underpinned by its dual action on mitigating oxidative stress and maintaining membrane integrity. Compared to the control, coated *T. fuciformis* exhibited significantly enhanced antioxidant capacity, manifested through higher activities of key antioxidant enzymes (SOD, CAT, APX), increased levels of non-enzymatic antioxidants (AsA, GSH), a greater superoxide anion (O_2_^−^) resistance capacity, and improved DPPH radical scavenging activity and reducing power. As a direct consequence, the accumulation of MDA, an indicator of lipid peroxidation, was markedly reduced in the coated samples. Concurrently, the coating effectively modulated membrane lipid metabolism by inhibiting the activities of detrimental enzymes such as lipase, LOX, and PLD. This enzymatic inhibition, coupled with reduced oxidative damage, preserved the structural integrity of the membrane, as demonstrated by the retention of key phospholipids (PC and PI), delayed accumulation of PA, and significantly lower cell membrane permeability.

In conclusion, the ε-PL/CTS composite coating significantly extended the storage life and maintained the quality of postharvest *T. fuciformis*. This was achieved by enhancing the *T. fuciformis*’s antioxidant defense system, thereby reducing ROS accumulation and oxidative damage, and by preserving cell membrane integrity through the regulation of lipid metabolism. These findings highlight the potential of ε-PL/CTS composite coatings as a promising, safe, and effective strategy for the postharvest preservation of *T. fuciformis* and potentially other perishable edible fungi.

## Figures and Tables

**Figure 1 ijms-26-07497-f001:**
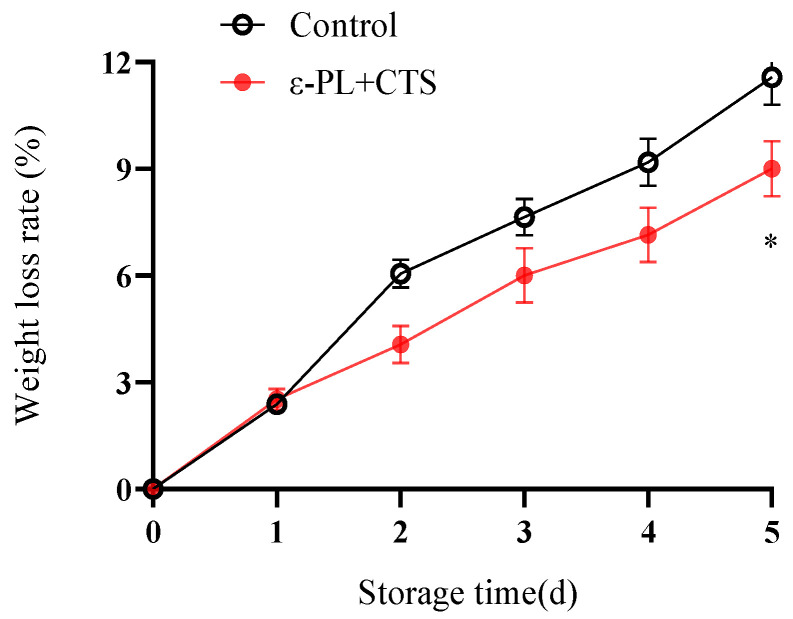
Effect of ε-PL + CTS composite coating on weight loss rate of *T. fuciformis* during storage at (25 ± 1) °C. Values are means ± SD (*n* = 3). Asterisks (*) indicate significant differences (*p* < 0.05) between the control group and the ε-PL + CTS group at the same storage time.

**Figure 2 ijms-26-07497-f002:**
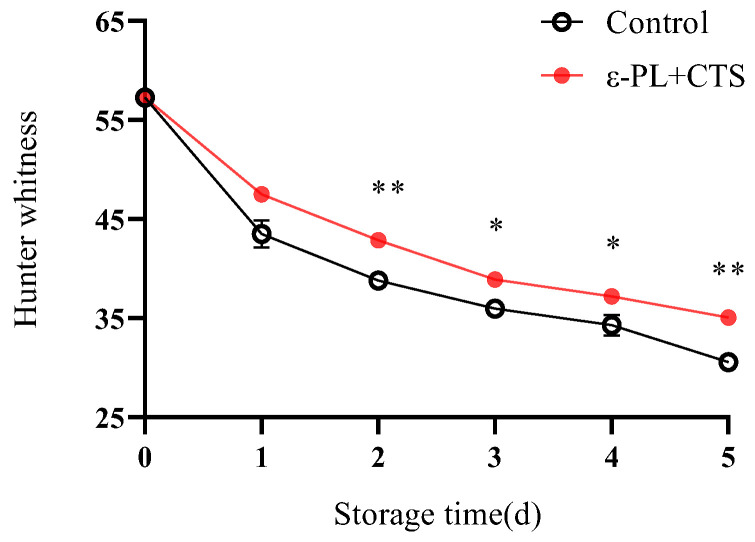
Effect of ε-PL + CTS composite coating on Hunter whiteness of *T. fuciformis* during storage at (25 ± 1) °C. Values are means ± SD (*n* = 3). Asterisks (*, **) indicate significant differences (*p* < 0.05 and *p* < 0.01, respectively) between the control group and the ε-PL + CTS group at the same storage time.

**Figure 3 ijms-26-07497-f003:**
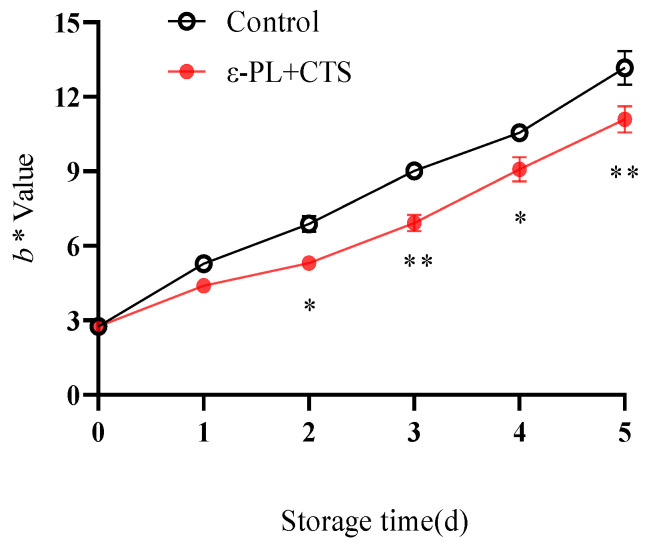
Effect of ε-PL + CTS composite coating on *b** value of *T. fuciformis* during storage at (25 ± 1) °C. Values are means ± SD (*n* = 3). Asterisks (*, **) indicate significant differences (*p* < 0.05 and *p* < 0.01, respectively) between the control group and the ε-PL + CTS group at the same storage time.

**Figure 4 ijms-26-07497-f004:**
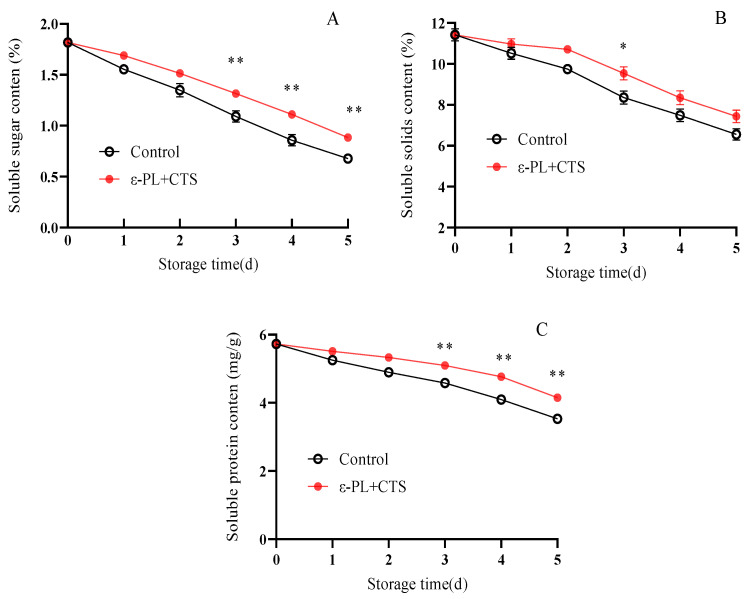
Effect of ε-PL + CTS composite coating on (**A**) soluble sugar content, (**B**) total soluble solid (TSS) content, and (**C**) soluble protein content of fresh *T. fuciformis* during storage at (25 ± 1) °C. Values are means ± SD (*n* = 3). For each parameter, asterisks (*, **) indicate significant differences (*p* < 0.05 and *p* < 0.01, respectively) between the control group and the ε-PL + CTS group at the same storage time.

**Figure 5 ijms-26-07497-f005:**
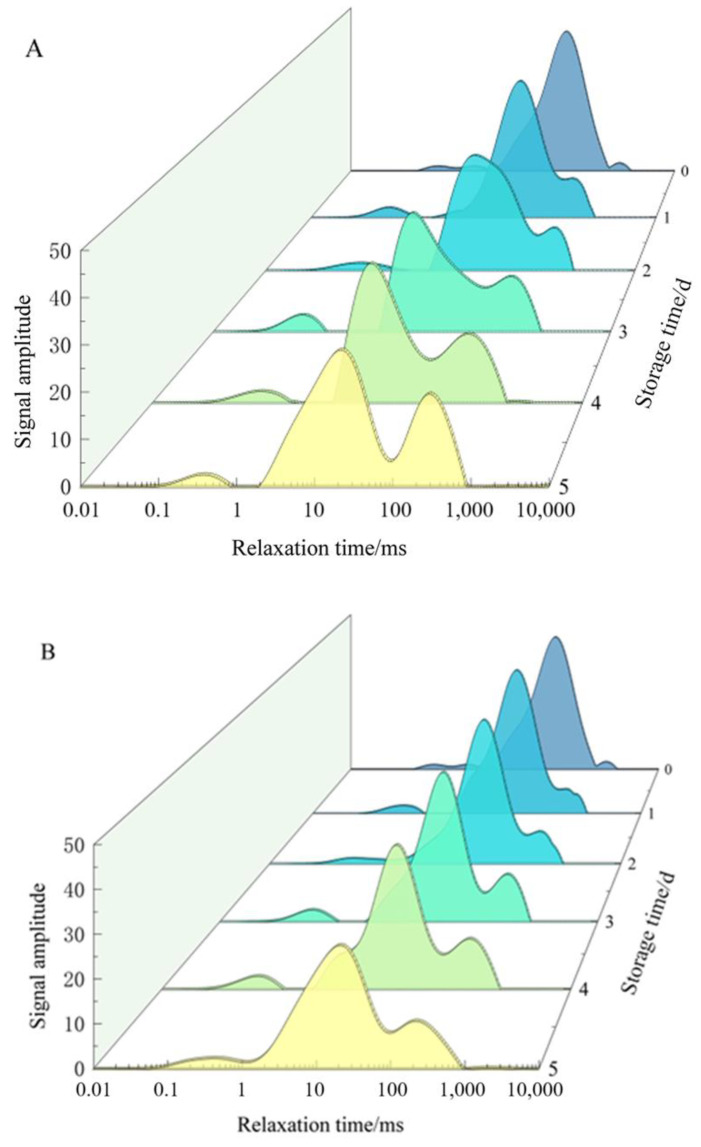
Effect of ε-PL + CTS composite coating on the lateral relaxation behavior of fresh *T. fuciformis* hydrogen protons. (**A**) is the control group; (**B**) is the ε-PL + CTS treatment group.

**Figure 6 ijms-26-07497-f006:**
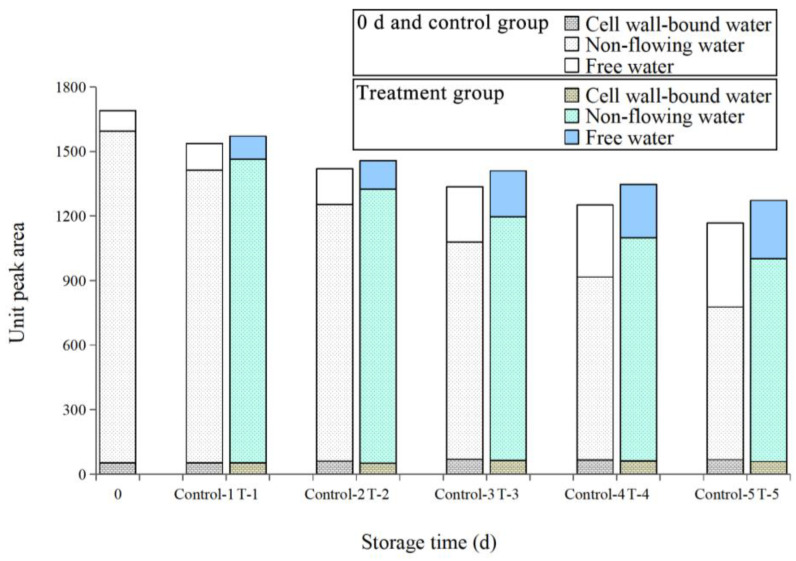
The impact of the ε-PL + CTS composite coating on the *T. fuciformis* unit peak area. The bar at day 0 represents the initial state before treatment. For days 1 through 5, “Control-X” and “T-X” denote the control group and the ε-PL/CTS treatment group on day X, respectively.

**Figure 7 ijms-26-07497-f007:**
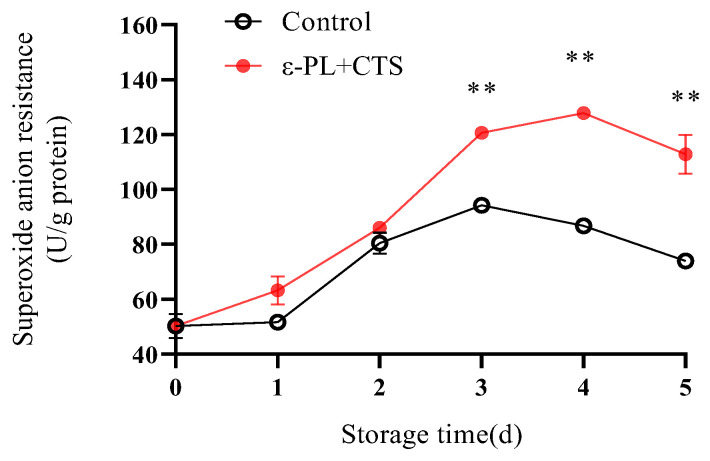
Effect of ε-PL + CTS composite coating on O_2_^−^ resistance capacity of *T. fuciformis* during storage at (25 ± 1) °C. Values are means ± SD (*n* = 3). Asterisks (**) indicate highly significant differences (*p* < 0.01) between the control group and the ε-PL + CTS group at the same storage time.

**Figure 8 ijms-26-07497-f008:**
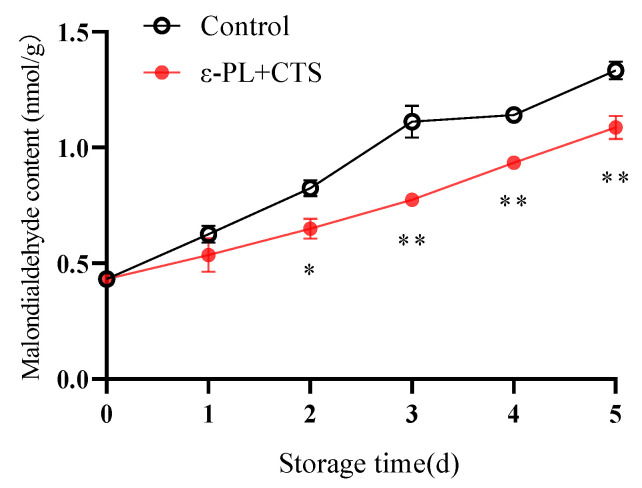
Effect of ε-PL + CTS composite coating on malondialdehyde (MDA) content of *T. fuciformis* during storage at (25 ± 1) °C. Values are means ± SD (*n* = 3). Asterisks (*, **) indicate significant differences (*p* < 0.05 and *p* < 0.01, respectively) between the control group and the ε-PL + CTS group at the same storage time.

**Figure 9 ijms-26-07497-f009:**
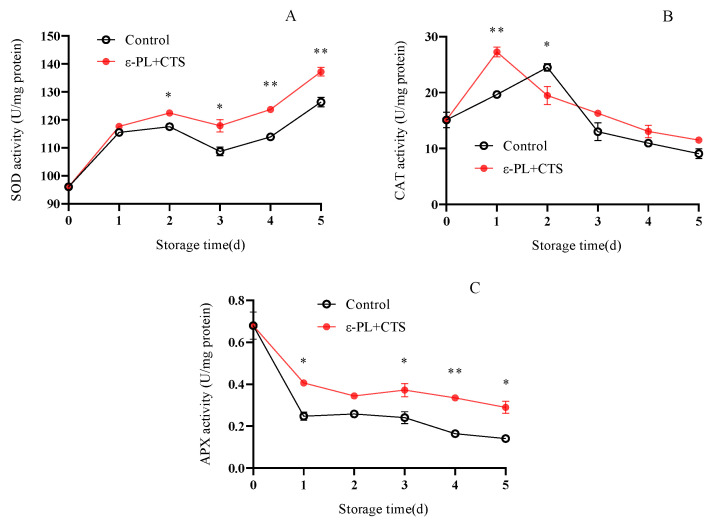
Effect of ε-PL + CTS composite coating on the activities of (**A**) superoxide dismutase (SOD), (**B**) catalase (CAT), and (**C**) ascorbate peroxidase (APX) in *T. fuciformis* during storage at (25 ± 1) °C. Values are means ± SD (*n* = 3). For each enzyme, asterisks (*, **) indicate significant differences (*p* < 0.05 and *p* < 0.01, respectively) between the control group and the ε-PL + CTS group at the same storage time.

**Figure 10 ijms-26-07497-f010:**
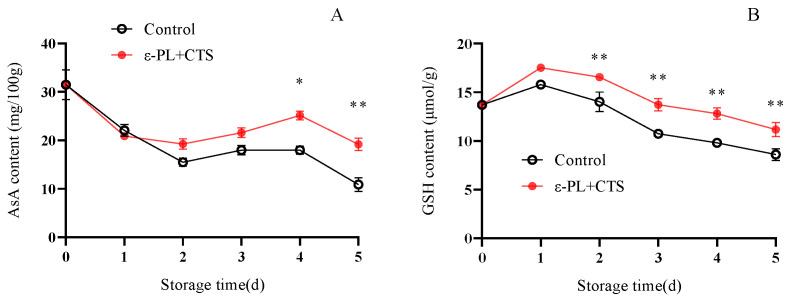
Effect of ε-PL + CTS composite coating on (**A**) ascorbic acid (AsA) content and (**B**) glutathione (GSH) content in *T. fuciformis* during storage at (25 ± 1) °C. Values are means ± SD (*n* = 3). For each parameter, asterisks (*, **) indicate significant differences (*p* < 0.05 and *p* < 0.01, respectively) between the control group and the ε-PL + CTS group at the same storage time.

**Figure 11 ijms-26-07497-f011:**
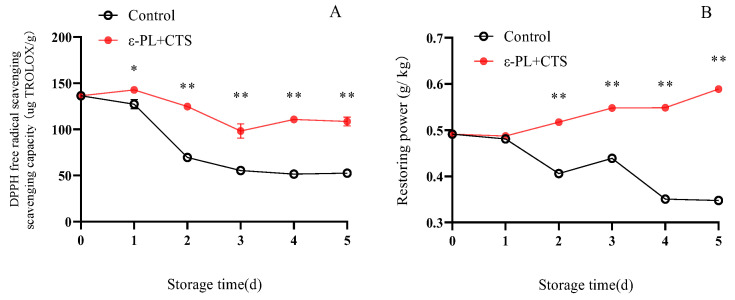
Effect of ε-PL + CTS composite coating on (**A**) DPPH radical scavenging capacity and (**B**) reducing power of *T. fuciformis* during storage at (25 ± 1) °C. Values are means ± SD (*n* = 3). For each parameter, asterisks (*, **) indicate significant differences (*p* < 0.05 and *p* < 0.01, respectively) between the control group and the ε-PL + CTS group at the same storage time.

**Figure 12 ijms-26-07497-f012:**
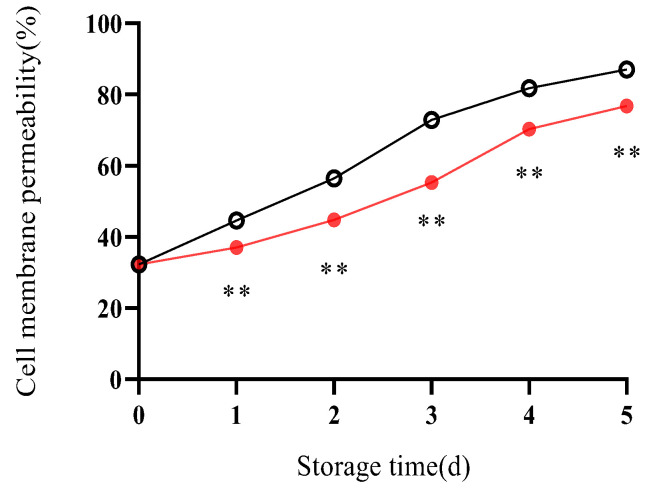
Effect of ε-PL + CTS composite coating on cell membrane permeability of *T. fuciformis* during storage at (25 ± 1) °C. Values are means ± SD (*n* = 3). Asterisks (**) indicate highly significant differences (*p* < 0.01) between the control group and the ε-PL + CTS group at the same storage time.

**Figure 13 ijms-26-07497-f013:**
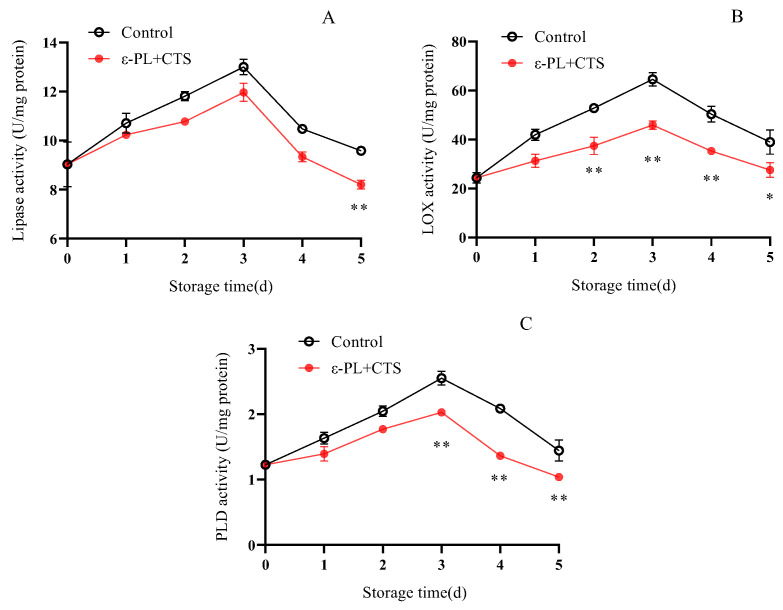
Effect of ε-PL + CTS composite coating on the activities of (**A**) lipase, (**B**) lipoxygenase (LOX), and (**C**) phospholipase D (PLD) in *T. fuciformis* during storage at (25 ± 1) °C. Values are means ± SD (*n* = 3). For each enzyme, asterisks (*, **) indicate significant differences (*p* < 0.05 and *p* < 0.01, respectively) between the control group and the ε-PL + CTS group at the same storage time.

**Figure 14 ijms-26-07497-f014:**
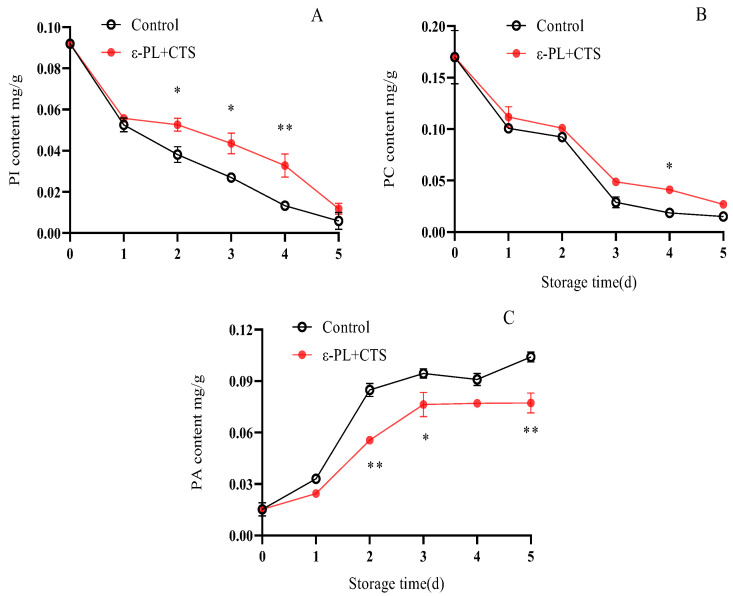
Effect of ε-PL + CTS composite coating on the contents of (**A**) phosphatidylinositol (PI), (**B**) phosphatidylcholine (PC), and (**C**) phosphatidic acid (PA) in *T. fuciformis* during storage at (25 ± 1) °C. Values are means ± SD (*n* = 3). For each phospholipid, asterisks (*, **) indicate significant differences (*p* < 0.05 and *p* < 0.01, respectively) between the control group and the ε-PL + CTS group at the same storage time.

**Table 1 ijms-26-07497-t001:** Effect of ε-PL + CTS composite coating treatment on the relaxation time of fresh *T. fuciformis*.

Treatment	Storage Time /d	Cell Wall-Bound Water T_21_/ms	Non-Flowing Water T_22_/ms	Free Water T_23_/ms
Control	0	0.767 ± 0.027 a	61.720 ± 10.606 a	414.227 ± 57.148 a
	1	0.341 ± 0.114 b	41.604 ± 2.884 b	383.673 ± 26.594 a
	2	0.279 ± 0.067 b	27.431 ± 1.902 c	431.171 ± 74.092 a
	3	0.254 ± 0.027 b	17.941 ± 7.588 c	333.933 ± 23.146 ab
	4	0.322 ± 0.068 b	10.777 ± 1.118 bc	209.368 ± 42.986 bc
	5	0.2945 ± 0.051 b	8.709 ± 0.302 c	126.038 ± 42.986 c
ε-PL + CTS treatment	0	0.767 ± 0.027 a	61.720 ± 10.606 a	414.227 ± 57.148 a
	1	0.391 ± 0.064 b	57.960 ± 3.171 a	398.673 ± 11.594 a
	2	0.301 ± 0.021 b	38.509 ± 9.177 ab	360.527 ± 49.740 a
	3	0.290 ± 0.010 b	29.970 ± 6.153 b	346.768 ± 35.981 a
	4	0.303 ± 0.042 b	22.703 ± 4.661 b	301.813 ± 31.317 ab
	5	0.389 ± 0.067 b	17.690 ± 3.040 b	212.273 ± 7.366 b

Note: Different letters after the relaxation time indicate the significance of the difference between different samples in the same column.

**Table 2 ijms-26-07497-t002:** *Bongkrekic acid* content (mg/kg) in *T. fuciformis* during storage at (25 ± 1) °C. ND: Not Detected (LOD = 0.015 mg/kg).

Treatment	Storage Time/d					
	0	1	2	3	4	5
Control	ND	ND	ND	ND	ND	ND
ε-PL + CTS treatment	ND	ND	ND	ND	ND	ND

Note: ND means *B. acid* was not detected in the sample.

## Data Availability

The original contributions presented in the study are included in the article, further inquiries can be directed to the corresponding author.

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
