# Peer review of "Effects of ε-Poly-L-Lysine/Chitosan Composite Coating on the Storage Quality, Reactive Oxygen Species Metabolism, and Membrane Lipid Metabolism of Tremella fuciformis"

_ijms, 2025, doi:10.3390/ijms26157497_

Round 1

Reviewer 1 Report

Comments and Suggestions for Authors

Dear Editor and Authors,

The manuscript investigates the effects of an ε-poly-L-lysine/chitosan (ε-PL/CTS) composite coating on the storage quality of Tremella fuciformis. While the study presents valuable insights into preservation mechanisms, several methodological and interpretive aspects require strengthening to enhance scientific rigor and translational relevance. Below are key concerns and suggestions for improvement:

  • Line 118, the study should include multiple concentration gradients of the ε-PL+CTS composite coating (e.g., 100 mg/L ε-PL + 3 g/L CTS, 200 mg/L ε-PL + 7 g/L CTS) to determine the optimal ratio and validate dose-dependent synergistic effects.
  • Lines 205-212,H2O2 ontent measurement should be added to ROS metabolism analysis, and membrane lipid metabolism should include fatty acid composition data (e.g., UPLC-MS) to clarify pathways.

3、 Line 315,ANOVA should supplement t-tests for multi-timepoint comparisons, with specification of variance homogeneity checks (e.g., Levene's test).

4、 Line 320,Direct comparisons with single CTS or ε-PL treatments should be included to highlight the composite coating's advantages.

5、  Line 436,The 3D waterfall plot in Figure 5 could be replaced with time-point 2D comparisons with labeled peak shifts for clearer water migration visualization.

6、 Lines 515-519,Quantitative microbial analysis (e.g., total aerobic bacteria, molds, yeasts) should be added to comprehensively assess antimicrobial efficacy, despite Bongkrekic acid being undetected.

7、The method for ensuring uniform coating application should be clarified (e.g., spray pressure, distance, coverage), or SEM could be used to observe coating distribution for reproducibility.

8、Additional experiments under refrigeration (e.g., 4℃) should be included to better reflect real-world cold chain logistics beyond the current 25℃ testing.

9、An Arrhenius equation or Weibull model based on key indicators could predict shelf-life extension under different temperatures.

10、Data on retention of key nutrients (e.g., polysaccharides, amino acids) should be provided to demonstrate functional property protection.

11、The impact of varying humidity levels (e.g., 60%, 90%) should be explored beyond the fixed 80% RH condition.

12、Compliance with food safety standards (e.g., GB 2760) or cited toxicological studies should confirm ε-PL+CTS safety.

13、The discussion should better link ROS and membrane lipid metabolism through signaling pathways (e.g., PLD+PA) while avoiding results redundancy.

14、Recent (2023-2025) high-impact food preservation literature should be cited to enhance novelty.

15、 The abstract should quantify preservation effects (e.g., "extends shelf life by X days") to strengthen practical value claims.

Reviewer 2 Report

Comments and Suggestions for Authors

The article has practical interest in the field of post-harvest preservation of mushrooms. The topic is relevant. A complex analysis of biochemical indicators demonstrates a systemic effect of the coating. The methodology of storage and quality control meets the standards. However, the work has significant comments.

Introduction:

The authors describe in great detail the value of the mushroom Tremella fuciformis and the region where it grows, as well as the antimicrobial and preservative properties of Chitosan (CTS) and ε-Poly-L-lysine (ε-PL). However, the authors write nothing about alternative preservation methods and their disadvantages, which the authors are trying to avoid.

The introduction does not reveal the novelty of the work and the advantages of the technology being developed. The literature review on the research topic is poorly developed.

Materials and methods:

The molecular weights of CTS and ε-PL and the degree of deacetylation of Chitosan are of great importance for the charge density, the stoichiometry of the complex during electrostatic association and the kinetics of formation of interpolyelectrolyte complexes. However, the authors did not indicate these important parameters.

Paragraph 2.4.9.1. – please indicate the full name of Kits.  

Results and discussion:

The authors write about the synergistic effect of CTS and ε-PL, but do not provide the dynamic effects from their own substances separately. It is necessary to clarify why these particular ratios of CTS and ε-PL were chosen.

There is no description of what the CK curve in the figures means.

In Fig. 3 - the authors claim a significant difference between the control group and the drug under study. However, a difference of 20% does not seem significant. Please, provide examples of other preservatives for this parameter.

Fig. 5 has low quality, making it difficult to analyze.

To make reading easier, it would be recommended to present a short preamble before each subchapter that reveals the importance of the study. Authors should provide a table with the main characteristics of the coating being developed and its analogues.

The authors need to confirm the formation of the complex "ε-PL and CTS and its synergism (FTIR, ζ-potential, which changes when the complex is formed).

No data on the thickness, homogeneity or adhesion of the film to the surface of the fungus, AFM/SEM

The work contains significant gaps in the fundamental substantiation of the mechanisms of action of the composite coating. The conclusions about "synergism" looks like speculative because the authors did not present an in-depth analysis of the physicochemical properties of the ε-PL/CTS system.

Round 2

Reviewer 1 Report

Comments and Suggestions for Authors

Accept

Comments on the Quality of English Language

None

Reviewer 2 Report

Comments and Suggestions for Authors

I still remain of the opinion that the research into the issues mentioned in comments 9 and 10 would greatly enrich the work. However, taking into account that the authors make significant reworked of this article and plans to perform additional experiment I can recommend this work for publication. Currently the article looks like holistic, its quality got much higher.